# The Lonely, Isolating, and Alienating Implications of Myalgic Encephalomyelitis/Chronic Fatigue Syndrome

**DOI:** 10.3390/healthcare8040413

**Published:** 2020-10-20

**Authors:** Samir Boulazreg, Ami Rokach

**Affiliations:** 1Faculty of Education, University of Western Ontario, London, ON N6A 3K7, Canada; 2Department of Psychology, York University, Toronto, ON M3J 1P3, Canada; arokach@yorku.ca

**Keywords:** myalgic encephalomyelitis, chronic fatigue syndrome, loneliness, psychosocial

## Abstract

This article provides a narrative review on myalgic encephalomyelitis/chronic fatigue syndrome (ME/CFS) through a psychosocial lens and examines how this impairment affects its sufferers during adolescence and adulthood, as well as how it impacts family caregivers and healthcare professionals’ mental health. Since there has been a lack of investigation in the literature, the primary psychosocial stressor that this review focuses on is loneliness. As such, and in an attempt to help establish a theoretical framework regarding how loneliness may impact ME/CFS, loneliness is comprehensively reviewed, and its relation to chronic illness is described. We conclude by discussing a variety of coping strategies that may be employed by ME/CFS individuals to address their loneliness. Future directions and ways with which the literature may investigate loneliness and ME/CFS are discussed.

## 1. The Lonely, Isolating, and Alienating Implications of ME/CFS

Myalgic encephalomyelitis/chronic fatigue syndrome (ME/CFS) is a debilitating neurological disorder known to produce a wide range of devastating symptoms best known to include extreme fatigue, pain, and post-exertional discomfort. Though it is thought to originate from a genetic predisposition and/or an interaction with a host of environmental factors (e.g., frequent injury), the exact precursors of this disorder are still not well-understood [1]. Researchers have, however, observed some uniformity in attempting to distinguish the properties of this illness; for example, patients with ME/CFS have been observed to have a reduced blood perfusion rate in the brain stem [2]. Additionally, abnormality in multiple brain structures that regulate pain has been observed [3], leading recent research studies to theorize that the ME/CFS brain’s homeostatic processes that react to pain are aberrant in nature [1]. This unusual brain activity, viewed plausibly due to a viral infection that impacts the central nervous system [4] such as, for example, glandular fever shortly before a diagnosis [5], has overarching effects that can cause cognitive difficulties, sleep dysfunction, and immune system irregularity, amongst other debilitating outcomes [6].

From the start, the conceptualization of ME/CFS as an illness has been riddled with controversies and dismissiveness from the medical community. Initially deemed to be psychosomatic in nature [7] and as epidemic hysteria [8,9], this mislabeling has persisted to this day, leading researchers to make observation that the medical community continues to harbor “prejudiced opinions that it is not a real illness” [10] (p. 309). One example of this was seen in Canada; in 2016, the federal government’s scientific panel rejected a grant application for ME/CFS research, implying that “it was not a disease” [11]. Though funding towards biomedical research devoted to researching ME/CFS was eventually accepted less than a year ago [11], this example provides an illustration of the ignorance faced by those living with the illness. Consequently, institutes such as the National Academies of Sciences Engineering and Medicine presently classify ME/CFS as a stigmatized illness [12].

While the ignorance directed towards this illness may be in large part be due to a holistic lack of scientific understanding surrounding the exact antecedents of this disorder [1] and the lack of a standardized procedure for determining its presence in patients [13], ME/CFS is known to cause irritation between the brain, the spinal cord, and the musculoskeletal system [14,15,16,17]. It is thus why the term myalgia encephalomyelitis was coined as the Latin word “myalgia” translates to muscle pain, “encephalo” to brain, and “myel” to spinal cord [18]. However, this understanding, as well as the incapacitating physical symptoms presented earlier, only tells one side of an ME/CFS sufferer’s story. That is, sufferers of this disorder typically also carry with them a myriad of negative non-physical consequences affecting anxiety, depression, and overall well-being [14]. Put another way, an adjustment period filled with many new expected and unexpected vulnerabilities occurs following a diagnosis of ME/CFS. Additionally, while variability exists in the way this syndrome affects individuals [8], for the most part, the major non-physical changes that ME/CFS patients commonly must endure include learning how to cope with psychosocial impairment related to the family structure, a loss of self, and a reduced social network [10,19,20,21,22]. 

As such, this review aims to explore these psychosocial implications as they relate to ME/CFS in adolescence and adulthood, as well as to highlight its impact on the caregivers and parents of patients/children with ME/CFS. In doing so, a predominant focus on the lonely, alienating, and isolating features of this illness is explored. It is important to note that though some researchers have found themes of loneliness when investigating persons with ME/CFS (e.g., [10,21,22,23,24]), the consequences of loneliness as it pertains exclusively to ME/CFS have not been studied. Additionally, while some researchers have found themes of loneliness in their research (e.g., [10]), the authors of these studies were not interested in the variable of loneliness from the outset. Given that loneliness has been observed to influence the expression of symptoms in chronic illness (e.g., having the ability to exacerbate symptoms in chronic illnesses—a topic which is later explored in further detail), we wish to attend to this omission in the literature. 

Thus, through a narrative review, this paper attempts to address multiple issues. It first aims to highlight the relevant psychosocial implications of ME/CFS; secondly, an overview of what loneliness is, how it relates to chronic illness (and illness in general), and its stigmatized connotations is presented; thirdly, this paper offers suggestions as to how to cope with loneliness stemming from and enhanced by chronic illness. This is done in hopes of constructing a theoretical framework for future research that wishes to bridge the gap in the literature between ME/CFS and loneliness.

## 2. Method

In making the assertion that no articles have directly investigated the impact of loneliness and ME/CFS, we conducted search queries that looked for ME/CFS keywords (i.e., “ME/CFS,” “CFS/ME,” “chronic fatigue syndrome,” and “myalgic encephalomyelitis”) and paired them with loneliness-related keywords (we used the terms “loneliness,” “lonely,” “isolation,” “isolated,” “alone,” “alienating,” and “alienation” as a possible pair to each of the ME/CFS keywords). Our search criteria, spanning 28 possible search queries, found no results on APA PsycNet, Google Scholar, and PubMed when requiring that at least one of each keyword be in the title of an article (e.g., a search query through google scholar was typed as the following: “allintitle: ME/CFS loneliness”). Next, we redid the search queries but, instead of requiring that a loneliness-related keyword be in the title, we changed this criterion to allow these words to appear anywhere in the abstract or in the body of an article.

Over 400 articles were researched. Of this, 137 studies were appropriate for our topics and were thus utilized. These articles mainly consisted of primary studies and textbooks; however, they also included literature reviews, meta-analyses, systematic-analyses, research instruments, and annual reviews. The search criteria were filtered between the years 1967 and 2020; however, a predominant emphasis was placed on articles within the last 10 years (of the total 137 studies, 18 dated between 1967 and 1999; 20 dated between 2000 and 2005; 23 dated between 2005 and 2009; 32 dated between 2010 and 2014, and 40 dated between 2015 and 2020).

## 3. Results

### 3.1. Psychosocial Factors of ME/CFS

The presence of chronic pain can have devastating effects on one’s psychosocial functioning. In fact, when it comes to chronic pain, some researchers consider the accompanying psychosocial distress to be so severe in magnitude that they advocate for a dual-diagnosis—one that includes a component on pain severity while also emphasizing the debilitating function of one’s social environment [25]. As Skelly and Walker [25] indicated, such a diagnosis would allow for the healthcare field to truly acknowledge “the way pain affects people’s lives, how they adjust to this, and how it affects their behavior” (p. 253).

For example, depression and anxiety are almost always comorbid in the presence of ME/CFS [26,27]—so much so that some researchers postulate that the term “comorbid” does provide an accurate picture. For instance, Maes’ [28] review of the neural pathways affected by ME/CFS and depression found that these two conditions do not exist independently of each other; instead, they are manifestations of similar damaged neural pathways. Thus, rather than using the term “comorbid,” Maes [28] advocates for the identification of ME/CFS and depression as “co-associated” disorders. This linguistic distinction, though small, may benefit the ME/CFS community through earlier mental health intervention, as healthcare professionals would be vigilant of depression from the outset of diagnosis.

#### 3.1.1. How ME/CFS Affects Sufferer’s Mental Health

A common reason why mental health issues are high with patients with ME/CFS is the frequent observance of kinesiophobia (i.e., a fear of movement, [29,30], which may cause withdrawal and isolation from one’s social circle due to the strenuous efforts of displacing oneself. Another common observance in ME/CFS individuals, which is devastating in tandem with kinesiophobia, is the act of catastrophizing [31,32].

Catastrophizing, defined as the general tendency to assume that the worst-case scenario will happen, presents challenges for the overly cautious ME/CFS sufferer. Pessimism, a fear of movement, and an intense irrational fear of expecting the worst to occur leads these individuals to isolate in an attempt to protect themselves from potentially negative exposure. However, this approach to confine and stay away from social and recreational pursuits is counterproductive; in fact, studies investigating the implications of social deprivation have shown that it instead increases pain perception [33,34]. Thus, kinesiophobia and catastrophizing may create a sort of negative feedback loop where an individual, wishing to mitigate symptoms, stays at home to “protect” themselves, only to have significant distress and an increase in pain.

While these issues may universally impact individuals with ME/CFS, the age-onset produces additional distinct circumstances that can affect psychosocial well-being in common ways. This is also true for caregivers of family members and patients of ME/CFS. As such, we devote the following section to explore further the different variations of ME/CFS’ psychosocial impairments and stressors.

#### 3.1.2. Psychosocial Implications of ME/CFS during Adolescence

While likely underestimated due to a lack of understanding and testing for ME/CFS, the prevalence of this diagnosis amongst teenagers varies between 0.11 and 1.9% [35,36]. Youth diagnosed with a chronic illness that damages the central nervous system (as ME/CFS has been observed to do [4]) are 40% at risk of a psychosocial impairment of some sort [37]. Additionally, compared to non-ME/CFS adolescents, teenagers battling this illness have shown disturbing social and emotional development, higher rates of depression, and a host of other negative implications related to school absenteeism [35].

Regarding school absenteeism, one study that included a sample of 81 adolescents with ME/CFS found that 45% percent of participants had missed a minimum of 50% of school during the previous six months. Such significant absenteeism lends itself to missed opportunities to develop peer relationships and social competence, future work-relevant skills (e.g., resilience and persistence), and academic and language development [38]. To make matters worse, the time missed from school does not even allow for optimal therapy to be had for these adolescents. For example, a study from the University of Bristol [39] found that only 10% of adolescents report having access to a specialist, and as much 94% of students report being disbelieved when disclosing their symptoms to health care professionals and school staff members.

Prolonged absence from school can also enhance a child’s feelings of isolation due to reductions in networking and participation in sports and other social events [40]. Adolescents are also especially prone to body distortion, and chronic illness can heavily disrupt body image. As Vitulano [40] states, “…bodily changes, and treatment requirements are nagging reminders that they are ‘different and damaged’ in some way” (p. 587), further impairing social involvement, acceptance, and self-esteem.

It is not surprising then that ME/CFS during adolescence is associated with a substantial loss of self. To illustrate, we would like to draw attention to Parslow et al.’s [21] study, one of the first major systematic reviews and meta-ethnographies of the ME/CFS qualitative literature during adolescence. Their findings indicated (after translating and coding 10 studies and 82 participants’ quotes on feelings of the syndrome) that the disruption and loss of self, alongside pain and social disturbances, was the most frequently emergent construct in the investigations reviewed. To further illustrate this, adolescents with ME/CFS were found to place their identities on their bodies; upon learning that their newfangled bodies limited their ability to behave as they used to, self-esteem and confidence, along with a loss of self, were negatively affected [10].

A loss in self may also be in part explained by the results of Winger et al.’s [22] qualitative study that found adolescents with ME/CFS attach significance to attending school and hang-outs with friends. When deprived of these events, the adolescent questions the meaning of life. Results from this study also found overwhelming feelings of loneliness and feelings of “sadness and guilt related to being a burden on one’s family” [22] (p. 2652).

Lastly, sibling rivalry was also observed to be an issue for adolescents with ME/CFS. More specifically, enviousness and abandonment are felt when siblings of ME/CFS sufferers go outside the home and live the otherwise “normal” adolescent experiences. Wiliams-Wilson’s [10] wrote about the effects of the illness, as experienced by Lisa, one of the participants: “Lisa believes that her having CFS has had a detrimental effect on her relationship with her sister; they argue frequently and are jealous of each other, they do not spend a lot of time together as her sister is either at school or the stables. Lisa mentions that she and her sister are jealous of each other, which is a point raised by other participants in the study” [10] (p. 200).

#### 3.1.3. Psychosocial Implications of ME/CFS during Adulthood

Chronic illnesses, in general, significantly affect one’s psyche. As Kiliçkaya and Karakaş [41] stated, “it is known that medical and psychosocial problems in chronic diseases cause negative emotions such as anger, distress, and unhappiness and that patients who have chronic illness feel loneliness. Hospital stays, taking medication, physical and social loss of function, economic setbacks, a changing body, and uneasiness in social relationships are factors that affect loneliness” (p. 486).

According to the World Health Organization’s investigation in 15 centers across the Americas, Europe, Asia, and Africa, chronic and persistent pain affects 22% of the population globally [42,43]. Of the affected population dealing with severe chronic illnesses such as ME/CFS, independence and autonomy are greatly impacted causing individuals, well into adulthood, to have an increased dependence on older siblings, parents, and/or caregivers for everyday functioning [40,44]. Subsequently, many adult sufferers are left with many unpleasant uncertainties when envisioning the future. For instance, patient testimonials reveal that a common concern for these adults includes fears of starting a family [25] (p. 189), which may develop into severe psychological trauma.

Economic setbacks and financial strain are further distressing side-effects for adults with ME/CFS. Many sufferers of the disorder are left too debilitated to travel to their office or work full-time, leaving them to seek fewer demanding jobs (with less pay) as a result [6]. Furthermore, in countries where patients have to pay for their medical bills (e.g., the United States), ME/CFS sufferers are expected to pay significant annual expenses related to treatment [45], leaving some with the illness to constantly worry and ruminate over how to clear any financial deficits. Thus, it is no wonder why a large body of literature reflects on the overall quality of life being significantly impacted as a result of ME/CFS [46].

This well-being is also affected by the disruptions caused to the overall routine of the ME/CFS sufferers. For instance, besides work and social habits, individual habits and the ability to partake in hobbies and recreational activities are also greatly impaired. To illustrate, Schweitzer et al. [47] found through a sample of 47 ME/CFS patients that 100% of respondents reported reduced recreational activities and 70% were forced to discontinue all physically active pastimes.

#### 3.1.4. Caring for Someone with ME/CFS

Due to the disabling nature of ME/CFS, there is often a need to heavily rely on someone to look after and help individuals dealing with this disorder [19]. The duties of caregivers and caretakers are often numerous in scope; as [48] Girgis et al. describe it, caring for people with a chronic illness includes, but is not limited to, assistance with “mobility, transportation, communication, housework, management and coordination of medical care, … emotional support, assisting with personal care, organizing appointments, social services, assistance with social activities, … and managing finances” (p. 197).

As a result, and independent of whether this role is fulfilled primarily by a loved one or by a health care professional, looking after and caring for individuals with ME/CFS can also produce significant psychosocial maladjustment for the caregivers/caretakers themselves. One study that focused on tracking family caregivers found that, on average, caregivers spend 13 hours per day assisting family members with ME/CFS, and this profoundly affected sleep, work and study time, leisure time, and mood [49]. Furthermore, this same study found that 63% of family caregivers described feeling depressed as the main stressor endured as a result of looking after someone with ME/CFS [49].

This emotional toll can also be expressed in different ways. For example, Catchpole and Garip [19] found that parents often face skepticism and disbelief by their social circles (i.e., individuals outside the immediate family, such as friends and colleagues), causing them to withdraw from these networks and experience increased isolation as a result. This behavior was rationalized to be a common coping mechanism for caregivers as their isolation effectively shielded them from the possibility of this type of criticism.

Additionally, both parental caregivers and professional caretakers reportedly endured significant mental health problems as a result of caring for ME/CFS suffers, and, even worse, this mental health burden was also related to a reduced ability to effectively care for the persons with ME/CFS [20,50].

Parents also experience role ambiguity when tending to children with ME/CFS, as the illness is said to create some confusion on how to approach the parent-child dynamic, which can lead to a strained relationship. This was reasoned to be due to the illness creating reduced opportunities to enjoy shared activities that otherwise promote bonding experiences and allow the parent and child to relate to each other [19]. Another example of this occurs when the amount of missed school starts to accumulate—since parents often homeschool their children in these instances and assume the role of teacher, the parent-child relationship may be affected due to the frustrations inherent in coursework and homeschooling [10].

Related to household tensions, Missen et al. [20], tracking 28 mothers of children with ME/CFS, also found marital tensions and problems within the marriage related to the disorder as the theme that was the “broadest and most widely discussed” (p. 5). Perhaps contributing to marital tensions, feelings of grief are also felt in ME/CFS households. Put powerfully, and somewhat pessimistically, parental caregivers “mourned the perceived loss of their child’s future” [19] (p. 5).

Another source of stress related to a child’s diagnosis with ME/CFS that we previously briefly alluded to, and is otherwise under-discussed in the literature, is sibling rivalry. Earlier, we explained this from the point of view of the adolescent sufferers. However, the siblings of sufferers also experience significant angst as a result of a diagnosis of ME/CFS in the household, and parents have been said to often ruminate about the impact this has on the siblings of the child sufferer [20]. For example, as one parent testimonial revealed in Missen et al.’s [20] study, “The main thing I worry about is [little sister] because she can’t now do the things that [child with ME/CFS] used to go out and do … unless I leave [child with ME/CFS] at home. So, I feel guilt about leaving [child with ME/CFS] and going out having a good time with [little sister]” (pp. 509–510). Additionally, the siblings themselves experience a wide range of stressors. Houtzager et al. [51] observed that the siblings of chronically ill children sometimes display even more emotional and behavioral turbulence than parents do during the initial adjustment period (i.e., when a chronic illness diagnosis is first given), as they find themselves having to rapidly cope with losses of attention and companionship from both the parents and the affected sibling.

In some situations, parents, after making sacrifices (e.g., reducing work hours and time spent with friends) can also develop forms of anxiety as a result of long-term homebound treatment administered to their child with ME/CFS. For example, Williams-Wilson [10] recounted, “one woman told me, outside of an interview, that she had been confined to home for such a long time that she actually felt slightly panicky on the rare occasions she visited a public place such as a supermarket, feeling overwhelmed by the number of people and amount of noise, she described it as having become institutionalised.” (pp. 252–253).

### 3.2. Loneliness

While the aforementioned psychosocial factors are numerous and greatly distressing, another equally troubling psychosocial impairment that individuals with ME/CFS must deal with exists—loneliness. Having so far provided a brief introduction to the ways in which this disorder may manifest itself detrimentally, we now do the same with loneliness by providing a brief background as to what it is, how it is expressed, and the stigma which is associated with it, before finally providing information on how it can affect chronic illness.

Loneliness, unlike solitude, which will soon be described, involves excruciating physical and mental suffering. Interestingly, we can find that the first thing that the biblical God named was loneliness, which is found to be associated with numerous somatic, psychosomatic, and emotional phenomena. Loneliness can be a reactive experience, that is aroused in response to a significant life change or loss, or it could be an essential experience which stems from one’s infancy and is intertwined in the individual’s personality [52]. Apparently, it was found that loneliness may have a significant or even profound impact on the brain and can affect reasoning, memory, hormone homeostasis, blood glucose levels, and one’s manner of addressing of physical and mental stresses and illnesses [53].

While various theories regarding loneliness have been advanced, several characteristics are unmistakably part of that experience: while in solitude, we choose to be alone in order to do what can be done only alone, e.g., reflecting, creating, sculpting, writing, taking a walk in the woods, or communing with nature. Loneliness, in contrast, is painful, unwanted, and difficult to tolerate. It thus motivates humans to seek meaning and connection. If we explore it from an evolutionary perspective, we can notice the manner in which animals survive and thrive. They can do so only when they are part of the herd, for the deer who lags behind will become lunch for the waiting lions. Thus, like physical pain, loneliness has an important survival function even though it is unpleasant (see also [54]).

Loneliness is an integral part of being human and is experienced in order to encourage us to connect and remain part of the community. Loneliness is an experience that includes cognitions, emotions, and behaviors that are mostly negative, turbulent, and unpleasant [55,56]. Loneliness is a universal experience; as a uniquely subjective experience, it results from a combination of the individual’s personality, social changes, and one’s history. That history includes, of course, the various experiences and illnesses with which one may have been afflicted [57].

We do not just require others for our survival and growth; we also particularly need the presence of those who support us, whom we trust, and with whom we can interact, work together, and prosper [54]. Thus, the mere physical presence of others is insufficient.

As humans, we need to feel connected to significant others. In general, the prevailing view is that being alone and perceiving oneself as unloved and uncared for will result in loneliness [58]. Research is heightening our awareness that in the Western Hemisphere, today’s fast-paced and constantly changing world where virtual reality can be seen as being on the brink of replacing the real one, people have little time and no energy to invest the effort required for establishing a connection with anyone beyond the narrow frame of their own hurried lives, living in and conforming to a culture that rewards nothing but the individual acquisition of power and money [59]. Cacioppo et al. [58] observed that “people are increasingly connected digitally, but the prevalence of loneliness (perceived social isolation) also appears to be rising. From a prevalence estimated to be 11–17% in the 1970s … loneliness has increased to over 40% in middle-aged and older adults … Over the past 40 years, loneliness has also become more widespread overseas” (p. 238) and is linked to poor physical and mental health outcomes.

At the time of this writing, COVID-19 has affected the entire globe, significantly changing the way we live. Dunham [60] indicated that loneliness could negatively affect the health of the brain as well as the immune system. This has also seemed to have been exacerbated by the confinement and lockdown. For instance, Fallon et al. [61] found through studying a sample of 431 individuals with chronic pain that they reported an increase in their pain perception and severity during the pandemic, which makes reviewing the effects of loneliness on those afflicted with ME/CFS even more poignant at this time.

#### 3.2.1. The Stigma of Loneliness

Most people are reluctant to admit, even to themselves, that they are lonely. Though we may geographically alone, feel unimportant, and unloved, people seem ashamed to acknowledge, let alone admit, that they are lonely. That is a consequence of the Western culture’s dictate that loneliness is a sign of weakness that should not affect “normal,” “healthy,” and “strong” people [62]. This denial does not eliminate loneliness; it simply conceals it from the world while we still hurt and feel alienated at times [63]. The increased use of drugs and alcohol, the purchasing and consumption of pornographic material, the very many calls to distress hotlines, and the rise in the number of suicides were found by research to be a consequence of the pain of loneliness that is not talked about and is not addressed. We can also see the footprints of loneliness in the increased number of divorces and religious fads. There is, clearly, a stigma to being lonely [64].

#### 3.2.2. Loneliness and Illness

Boehm [65] pointed to the connection of our emotional well-being, our thoughts, emotions, and behaviors to our well-being (see also [66]):

“*Individuals who are satisfied with their lives and who experience frequent positive emotions—that is, individuals with high levels of subjective well-being … —not only feel good but may also have reduced risk for developing coronary heart disease … subjective well-being may buffer against the harmful health consequences of stress and exert direct influence on bodily systems or may motivate healthy behavior*”(p. 1).

Modern medical science has been obsessed with death, which is a clear “enemy” in medical eyes, and, thus, medical research aims to eradicate the diseases that cause it [67]. Primary-care physicians consequently focus on the care of the patient and many times are not fully aware of the person who has the disease.

Illness is stressful and often frightening—even more to those who are physically disabled, immobile, or are close to the end of their journey on earth. Illness, in general, is a major stressor in one’s life [68]. Fatigue, pain, or, in severe cases, immobility results in the body being in a continuous state of stress. Such a situation leads to hospitalization and may cause a wide range of short-term and long-term negative effects experienced by the patient [69]. In general, physical suffering and distress plunge the body into a state of continuous stress that may be exacerbated by the patients’ negative psychological states. Such negative psychological states compound the patient’s stress and may result in a perception that one’s life is under threat, further suggesting that the illness is an uncontrollable or even unpredictable part of one’s life [70].

Stress, including separation, loss, and feelings of hopelessness are known to compromise the immune system and can reduce the body’s efficacy in fighting illness. Health deterioration is, thus, most probable in persons with already compromised immune functioning [71]. Loneliness, which is associated with a wide range of health problems, is linked to heightened morbidity and mortality. A positive correlation was found between social isolation and mortality [72]; a report in Australia by The National Heart Foundation reported strong evidence that social isolation contributes to coronary heart disease [73]. Loneliness has also been implicated in a lower level of quality of life [74].

The chronically lonely display negative mood, tend to withdraw socially, lack trust in others, and often are dissatisfied with their relationships [75]. Those with high loneliness tend to have poorer T-lymphocyte responses and show potentially harmful changes in natural killer cell activity [76]. Natural killer cells have a role in some cancers and inflammatory responses that have been observed in vascular disease [77].

#### 3.2.3. Illness Conceptualization

This section provides a brief overview of how people conceptualize illness as it may clarify our understanding of why loneliness is such an influential experience in the progression of illness. As Leventhal et al. [78] found, the following components relate to illness conceptualization.

The disease’s identity and label significantly influence patient behavior. For example, chest pain may be labeled “heartburn” and that will cause a very different behavioral reaction than the one labeled “heart attack.” Similarly, when the illness indicates a minor physical problem, we can expect that less emotional arousal will be experienced than if it is of a more serious nature.

Diagnosis may not always concur with timelines. For instance, people diagnosed with hypertension may view it as acute (although it is a chronic condition), and that has a direct effect on how much they adhere to treatment because it significantly differs from the way they may address chronic illness.

After diagnosis, we search for the cause of the problem. The cause may intimate that we need to seek treatment for it and, moreover, may influence the degree of our compliance with instructions given by a healthcare professional. For instance, pain in our leg that resulted from a fall would generate a completely different reaction than if it was found to indicate bone cancer.

The consequences of the disease form the next component. For instance, cancer may be viewed as a death sentence and result in the patient feeling hopeless and consequently failing to seek active and lifesaving treatment.

The degree of controlling the disease is the final component. If patients perceive that the situation is beyond hope, they may not seek treatment. However, if they believe that the treatment can help or even cure them, they will actively and even aggressively seek to achieve healing.

Research has also shown that stronger immune systems are positively associated with stronger social support systems [79,80], whereas people who have fewer social ties are more susceptible to illnesses [71]. Those with a solid social support network commonly cope better with stress and chronic pain [81], have better health, and have lower rates of mortality [82]. The nature of people’s connection to the community, and their perceptions of those relationships, significantly affect their physical and mental health [58].

What is even more staggering is that social isolation and loneliness rivaled cigarette smoking, high blood pressure, obesity, and a sedentary lifestyle as related to illnesses. Research has found a positive correlation between social support and health. Conversely, the opposite is also true. That is, those with the fewest social ties are up to four times more likely to die from illness and disease than those who had a good support system [54,81].

Segrin et al. [83] suggested that the lonely are less prone to behave in a health-promoting way, partly since they are not supported by others to adhere to a healthier lifestyle, which end up further increasing their chances to suffer from health problems [84]. Stress lowers the efficiency of the immune system, and loneliness can be a major stressor and contribute to ill health [85].

In sum, social relations seem able to protect us against the ill effects of stress, and those who lack social support end up with a greater allostatic load [83]. Loneliness may not only bring about illness, but it is known to corrupt the recuperative process [83,85].

#### 3.2.4. Loneliness, Chronic Illness and Pain

Chronic pain is quite pervasive and is estimated to affect 36%, or about 120 million individuals, in the United States [43,86]. Morrissey [87], focusing on the illness and suffering of older adults, highlighted the negative effect chronic pain has on the quality of life of all sufferers and highlighting how this causes a focus in attention on the losses that come as a result of chronic illness or pain. Pain also affects our psychological well-being by making us focus on persistent thoughts and irrational beliefs (such as the earlier mentioned kinesiophobia) related to individual reactions to the experience of pain or illness [88,89]. Recent research has indicated that expectations, mood, and behavioral factors also affect chronic pain; this by itself can significantly affect a person’s close relationships and social life [90]. Social isolation is also a major issue confronting chronic pain patients (Newton et al., 2013 [91]) and, thus, a strong association between chronic illness and pain and loneliness, as well as other emotions, has been found (e.g., [92,93]) 

Kool and Geenen [94] found, by comparing patients with fibromyalgia and rheumatic diseases, that patients with fibromyalgia were lonelier than those afflicted with rheumatoid arthritis. The same was found in a study on those with sickle cell disease [95]. High levels of social withdrawal and isolation were found in patients with neuropathic pain as they reported much social withdrawal and consequent isolation, and this, naturally, had an effect on both the patients and their spouses [96]. Loneliness was a major risk factor for the development of fatigue and depression in those patients [97], and social support and involvement have been found to be positively related to coping with pain [98,99].

Intrafamilial relationships can be a major source of personal resourcefulness for patients with chronic illnesses or pain. Family constellation can significantly impact the trajectory of chronic illness [100]. Research has repeatedly demonstrated a robust directional effect of loneliness on physical health across the lifespan [101].

Chronic illness may cause a loss of friends or family members and may thus intensify the loneliness that the ill person already experiences [102]. Loneliness is reported by patients who are forced to focus on their illness while the rest of the world, their family, and friends continue with their daily living [103]. “Individuals with high versus low chronic interpersonal stress were especially vulnerable to the negative effects of episodes of loneliness, showing greater loneliness-induced increases in cortisol… Beyond its physiological effects, one day’s increase in loneliness has been associated with increases in the next day’s symptoms, including exhaustion and fatigue, over and above the influence of the prior day’s depressed affect and sleep duration” [104] (pp. 929–930). Further, Wolf and Davis [100] asserted that physical pain and perceived social exclusion (which we term loneliness) activate brain circuits in the central nervous system where there may be a “pain signature” that is interestingly activated by either a physical or social stimuli [105].

ME/CFS sufferers experience profound fatigue, exhaustion, the loss of muscle power, pain, joint tenderness, and cognitive dysfunction. In addition, these stressful symptoms cause headaches, sore throats, a loss of concentration, and short-term memory loss [106]. It is quite clear that being riddled with such symptoms for a lengthy period of time would make socializing, interpersonal connection, and remaining connected to others problematic and would most often require a termination of those relationships. Everyday activities become burdensome for people with ME/CFS. They often lose the ability to keep up with a conversation since they experience extreme trouble focusing on what the other person is saying and, moreover, processing the meaning of the words. ME/CFS patients have trouble not only processing information but also retaining it. Memory loss, particularly short-term memory loss, is another common cognitive complication of ME/CFS. Sufferers forget people’s names, and that makes relating to them that much more difficult [107].

### 3.3. Coping with ME/CFS Induced Loneliness

As Biordi [108] pointed out, social isolation is a major aspect of chronic illness due to its significant impact on the patient and his or her support network. A variety of interventions, from high-touch and no-technology to low-touch and high technology use, have been suggested and tried. Here, we review a number of the main ones.

#### 3.3.1. The Power of Empathy

Bharadvaj [109] rightfully observed that any chronic disabling condition (and especially ME/CFS) can make a person feel challenged, anxious, or even hopeless; all of which are closely related to energy levels and healthy functioning of the immune and neuroendocrine systems. Low energy levels may cause the suppression of the immune system and the imbalance of the hormonal pathways. Consequently, Bharadvaj [19] observed a strong association between ME/CFS and mental health issues. He concluded by suggesting that “perhaps the best support a healthcare provider can offer is empathy and understanding to an individual suffering from ME/CFS. From a place of trust and rapport between doctor and patient, communication can begin about diagnostic testing, therapeutic options, and follow-up care. Just as important is the individual’s desire and hope in achieving wellness through lifestyle changes, psychological support, natural medicines, and anything else needed for the evolution in their health” (p. 92).

#### 3.3.2. Keeping in Contact with the Outside World

Feeling isolated can be a very common problem for people with ME/CFS. Being socially connected is a basic human need. Being ill may not allow one to get together with friends, but, as suggested by Campling and Sharpe [26], one can socialize in a different way. For instance, instead of going out, you may wish to invite some friends to your home for a meal you ordered or prepared yourself. When one is struggling with illness, some friends drop away as they cannot cope with that person’s illness. Making new social contacts and replacing those friends can address that problem, and leave the patient connected with the outside community.

#### 3.3.3. Peer Counseling

Whether it is informal in structure or more formal, initiated, and supported by the professional healthcare worker, Riegel and Carlson [110] suggested that peer counseling can be quite effective and helpful. For instance, a telephone hotline can be set up at a clinic that helped peers befriend each other and enable them to provide emotional support and active listening over the phone. A peer counselor volunteer may also be able to visit clients at home or in institutions and offer more social contact. Peers can also provide a wealth of information on ways to connect with resources such as assisted transportation, volunteers, friendly or financial aid [111]. This may also take form in the presence of support groups.

#### 3.3.4. Support Groups

A multitude of support and self-help groups exist for people in the general population, as well for those struggling with a debilitating illness. Research has indicated that support groups are very effective in meeting patients’ social needs by allowing them to exchange information, offer mutual support, learn of ways to cope with what they are going through, and ease their physical and emotional pain [108,112,113]. The internet can direct the person to sites related to the chronic illness that he is battling, and some associations have both national and local support groups according to not only to the patient’s illness but also their locations and ability to be mobile.

In addition, there may be ME/CFS support groups that are within reach of the patient where someone can get support and share their personalized experience on what it is to live with ME/CFS to the differently abled crowd. As we learn now, socializing does not have to happen face to face. Pen pals, friends, and telephone contacts can be very uplifting, as can emails, Facebook messages, etc. [114]. Brigden et al. [115] studied how adolescents coped with ME/CFS and noted that since the social connection was so important to them, they relied on the internet to connect to others and created a “community” of those suffering from the same illness, easing their isolation. They stated that in general, “the online world was less demanding and more flexible than offline relationships, especially in the context of a disabling and fluctuating illness” (p. 4) Additionally, one of their participants remarked that “It’s just the support knowing that at any time during the day if I’m having a bad day I can literally go on and I know immediately I’ll have support” (p. 4). Thus, it is suggested that whoever can gain access to the internet (and not just adolescents) may benefit similarly.

Some support groups may also be beneficial by offering physical activity. For example, Broadbent et al. [116] studied the use of aquatic exercise classes with ME/CFS patients for five weeks, offered in a biweekly manner. Besides the aerobic benefits, the results indicated that participants felt reduced social isolation and felt supported by their ME/CFS peers and exercise instructors, resulting in a reduction of pain, fatigue, and anxiety levels during post-treatment interviews [116].

#### 3.3.5. Solitude

A person’s choice to seek solitude is healthy. When it is chosen voluntarily, solitude is used for reflecting, centering, feeling spiritually connected, and finding inner peace and strength [117]. Solitude allows us to take a respite from everyday stresses, stimuli, and demands, and it also affords a better understanding of who we are, what do we want, and possibly how to get it [52]. ME/CFS sufferers may therefore benefit from a perspective shift on the extra time they have to themselves when being home-bound or otherwise away from their regular duties. Emphasizing personal growth may make these long bouts of aloneness more tolerable.

#### 3.3.6. A Cognitive-Behavioral Approach to Illness 

Campling and Sharpe [26] opined that ME/CFS patients who take control of their situation and actively attempt to help themselves are better able to overcome the emotional toll of their illness. On the other hand, those who believe that their symptoms are very severe, are caused by factors outside of themselves, and that they are “helpless” seem to be associated with greater disability. People relate to and are influenced by the people about them. While the research relating to this approach to coping is limited, we know that these social factors influence the degree to which an ill person struggles with ME/CFS.

Campling and Sharpe [26] indicated that pain which ME/CFS patients suffer from is made worse by muscle tension and, at times, is even caused by it. Consequently, they suggested that learning deep muscle relaxation may help reduce tension and ease the pain. They added that changing the way patients think and adopting a positive outlook will affect not only their mood and behavior but also their physiological state. They observed that “persistently inaccurate thinking can lead to poor coping and to bad effects on emotion, behavior, and bodily state. For instance, if a person constantly worries that things will go wrong, they will be chronically anxious, tend to avoid doing things, and be in a physiological state of tension and arousal” (p. 166). That will, in turn, affect the illness trajectory and their chances to recover. That may also influence them when they contemplate seeking treatment and their choice of preferred intervention, and it may further affect their nervous and immune systems.

To wit, we would like to re-emphasize the study by Fallon et al., [61] that was conducted in the midst of the COVID-19 lockdown in the UK (mid-April to early May 2020), where it was found that people with chronic pain reported self-perceived increases in levels of pain severity compared to the period before lockdown. The lockdown affected them more adversely than it affected the general population. They also reported greater increases in anxiety and depression, increased loneliness, and reduced levels of physical exercise. Evidently, the way the mind perceives one’s illness is a key contributor to that individual’s phenomenological experiences.

As such, Campling and Sharpe [26]) recommend cognitive behavioral therapy (CBT) to examine one’s thoughts, enhance rational thinking, and encourage positive and proactive cognitions which will usher similarly proactive behaviors. “Does CBT work for people with ME/CFS? [they asked]. Yes, it does seem to help. It is not a cure, but research including a number of clinical trials in different centers has shown that about two-thirds of patients who take part in such a program are able to do more and feel better” (p. 169).

#### 3.3.7. Religion

Research suggests that people rely on religion, spirituality, and faith to cope with illness and loneliness [118]. A study by Han and Richardson [119] identified spirituality as a coping strategy used to lessen loneliness in their sample of homebound elders. Religion was also found by Rokach [52] to assist in coping with loneliness in his research of both ill and non-ill samples. Being part of a religious community, as well as relying on one’s faith that a higher power is overlooking one’s life and suffering, was shown to ease suffering and help cope with loneliness.

#### 3.3.8. Spirituality

Spirituality is known to be a source of strength for many people. As we clarified previously, spirituality and religiosity are not the same. Spirituality can be experienced regardless of a person’s religious beliefs. Spirituality can promote the client’s feelings of control, self-esteem, meaning, and purpose in life. Nurses, with compassionate listening and sharing, when accompanying the ill can teach and enhance spirituality in the patient. Among the practices that may enhance spirituality are meditation, reading, yoga, tai chi, pet therapy, journaling, listening to relaxing and pleasant music, and repeating a mantra [117,120].

Spirituality achieved through mindfulness and meditation-based practices may also prove beneficial. For example, Boellingus et al. [121] found that mindfulness-based interventions and loving-kindness meditation can produce increases in self-compassion. In turn, self-compassion was seen to correlate to better day-to-day functioning in chronically ill patients. More specifically, its presence decreased pain perception and depression symptoms and increased work and social adjustment [122].

While spirituality can both mean and be achieved in different ways from individual to individual (e.g., meditation, mindfulness, and acts of gratitude), the main core aiding agent from these acts, besides the ability to relax, is the ensuing feeling of control. As Friedberg [6] stated, “studies on coping in ME/CFS and FM have found that a sense of control over symptoms consistently predicts better functioning, regardless of limitations or disabilities.” (p. 38) (see [123,124]).

#### 3.3.9. Health Care Providers Therapeutic Use of Self

In an article directed at healthcare professionals, Holley [117] suggested that nurses can be a major source of social support in their patients’ lives. Nurses are commonly perceived as trustworthy, compassionate, and knowledgeable, and they may even serve as confidants in many cases since they may be proficient in active listening. Even if the nurse cannot increase the patient’s social circle, he or she can provide caring, genuineness, and high-quality contact. Biordi [108] highlighted the authentic intimacy a patient and nurse can share and pointed out that their relationship may be a powerful one. Validating a patient’s importance as a human being, Biordi [108] suggested, can be as simple as stopping, making eye contact, and gently squeezing his or her hand.

As a way of summarizing the literature discussed thus far, Figure 1 provides a graphical recapitulation of the way ME/CFS and loneliness may interact through the different coping mechanisms and through the previously mentioned exacerbating effects.

## 4. Discussion

### 4.1. Summary

To reiterate, chronic illnesses that affect the central nervous system such as ME/CFS does produce psychosocial impairment in 40% of individuals [37]. Though this impairment varies from individual to individual, the age at which sufferers deal with the illness dictates broader and commonly reported themes. For example, adolescents plagued with this disorder miss significant amounts of schooling, which impedes social functioning and future career development skills and can lead to a loss of identity, all of which make young ME/CFS individuals question the meaning of life.

Additionally, a family who receives a diagnosis of ME/CFS for one of its members may experience disruptions of the family dynamic including sibling jealousies and rivalries, guilt, and, strained parent-child relationships resulting from parents and children needing to step into differing roles when assisting (e.g., a parent taking on the role of teacher when homeschooling or a child taking on the role of a parental figure when advising recommendations on what not to do). ME/CFS suffers who are single with no guardians and no dependents also have their own shares of concerns that they must deal with. This includes rumination and stress related to the financial impact of the disorder (e.g., the loss of work and the cost of treatment), fear about being unable to live a normal life and start a family, and decreased autonomy and an increase in reliance on a caregiver. Furthermore, the stigmatization of this illness results in dismissiveness and skepticism from peers, from authority figures (e.g., teachers and employers), and sometimes even from family members.

Concerning loneliness, the main focus of this article, we have provided a brief explanation of what loneliness is, how it may result in distress, unhealthy coping behaviors, and how it relates to chronic illness. In doing so, we have highlighted Leventhal et al.’s [78] study, which showed how one’s conceptualization of illness—e.g., the labeling of the illness and the perception of it, control over how one feels about it, expected consequences, and level of hopefulness—can greatly aid or vastly worsen one’s experience with their illness. Several coping strategies that caregivers and sufferers of ME/CFS may benefit from were also mentioned, including empathetic behaviors, the attempt to stay in touch with the outside world, peer counseling, support groups, solitude, and the cognitive-behavior approach to how to think about the illness. Additionally, we emphasized the important role healthcare professionals can have with their patients and spoke about the power of spirituality and religiousness as a buffer to ME/CFS-induced loneliness.

### 4.2. Future Directions

As we previously mentioned at the beginning of this article, there is a lack of investigation surrounding loneliness and how it affects individuals with ME/CFS. As such, we would like to raise some questions that would be of interest and offer insights into conducting research studies with this population.

Questions that glaringly present themselves are: can adequately managed and prolonged exposure to social support networks mitigate symptoms of pain in ME/CFS patients? Additionally, would being in a support group amongst other ME/CFS individuals offer the same buffers to loneliness non-ME/CFS groups? Might these effects be observable via online support groups (e.g., Zoom, Skype, etc.) and would they produce similar outcomes as in-person groups?

The length of illness and how it relates to loneliness are also of interest. For example, since ME/CFS symptoms are present for a minimum of six months and up to, in some cases, more than two years [21], a longitudinal study that tracks loneliness and how one perceives their diagnosis of ME/CFS (including pain, irritability, feelings of control) would be of great interest and could afford insight on whether or not lesser amounts of loneliness translate to a shorter length of pronounced distress faced by the illness. A specific look at personality traits, such as extraversion and introversion, and questionnaires related to perceptions of joy derived from outings, past job experiences/hobby enjoyment (e.g., Quality of Life Enjoyment and Satisfaction Questionnaire [125], The Minnesota Satisfaction Questionnaire [126], etc.) should also be noted and looked at for further perspectives on illness perception. For example, Davey et al. [122] found that individuals who ranked higher on openness to experience were more accepting of their own inner experiences dealing with chronic illness, resulting in significantly lower pain perception.

Additionally, while difficult, it would be fruitful to sample a comprehensive sample that includes many different cultures and/or backgrounds. Since different cultures are affected and tend to view loneliness differently [127,128,129], it would be interesting to observe if and how these cultural differences fare with respect to coping with ME/CFS. Answers to these questions would undoubtedly result in better treatment protocols and healthcare expectations.

## 5. Conclusions

Considering that loneliness, its accompanying stigma, and illness conceptualization have a devastating impact in exacerbating chronic illness, we deem the current lack of investigation between loneliness and ME/CFS a major omission in the ME/CFS literature. In closing, we wish to end this article on a quote from Williams-Wilson [10], a researcher who suffers from ME/CFS herself and who investigated the qualitative experiences of adolescents with ME/CFS; drawing from one of the emergent themes of her study, and her personal experiences, she remarked, “finding other people in the same situation as you, with the same struggles and daily trials makes one feel less alone and different from the rest of the world; it provides a sense of affinity and justification and helps alleviate feelings of isolation and loneliness.” (p. 317). It is thus a healthcare imperative that we take the necessary steps to study and demystify the illness’ alienating and isolating aspects so that those suffering with ME/CFS can feel empowered and compassion from the medical community when dealing with the disorder. Future research may explore the assistance that others, family members, friends, and the community at large can offer those who are struggling with ME/CFS loneliness-related stress and emotional pain.

## Figures and Tables

**Figure 1 healthcare-08-00413-f001:**
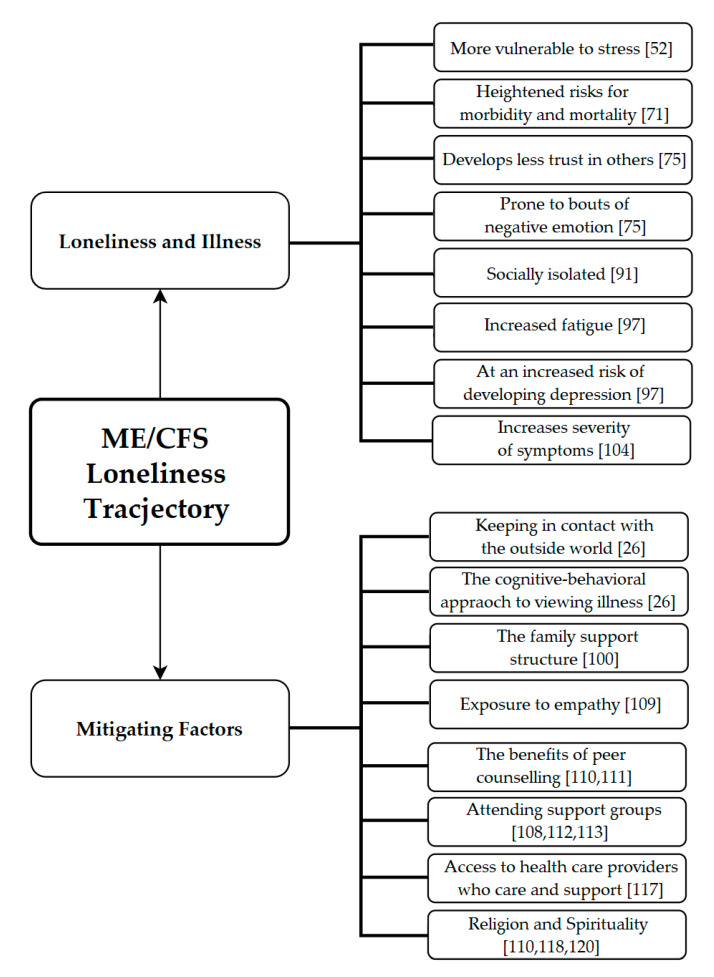
The interaction between ME/CFS and loneliness. [26,52,71,75,91,97,100,104,109,110,111,112,113,117,118,120].

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
