# Peer review of "The Lonely, Isolating, and Alienating Implications of Myalgic Encephalomyelitis/Chronic Fatigue Syndrome"

_healthcare, 2020, doi:10.3390/healthcare8040413_

Round 1

Reviewer 1 Report

I started to read the manuscript and the the second paragraph starts: From its conception, the conceptualization of ME/CFS illness has been riddled with 26 controversies and dismissiveness from the medical community. Initially deemed to be psychosomatic in nature and explained as epidemic hysteria (Friedberg et al., 2012; McEvedy & Beard, 1970; Neu et 28 al., 2014), this approach has persisted to this day, as researchers have observed members of the 29 medical community continue to harbor “prejudiced opinions that it is not a real illness” (Williams-30 Wilson, 2009, p. 309).

I checked the references because the hysteria concept was certainly initiated by McEvedy in 1970 but the other references have nothing to do with hysteria. This type of misquoting references occurs in other parts of the manuscript. The manuscript is not well written and shows a high bias in writing and is not giving a balanced assessment of the evidence.

Author Response

As per the reviewer’s suggestion, we now included a Method section which outlines how we went about our literature search. Additionally, we added a flowchart (on page 22) which clearly represents the material in a clear and easy to follow diagram.

As you will see we overhauled the manuscript, and revised the section on loneliness and on loneliness and illness.

We were unable to add a ‘Results’ sections, since this is a review and not an experiment. However, as per your suggestion, we added a Discussion section, and harmonized it.

Reviewer 2 Report

Authors reported well written and logically organized narrative review regarding Myalgic Encephalomyelitis/Chronic Fatigue Syndrome. This is undefined disease with a wide range of signs and symptoms predominantly categorized as pain, fatique, and phycological disorders. Authors thoroughly described how loneliness impacted on nature evolution of Myalgic Encephalomyelitis/Chronic Fatigue Syndrome and tied pervasive social isolation due to COVID-19 to the disease. This is interesting article, while authors used live spoken language instead academical approach in many cases. However, I congratulate authors on the narrative review. I do not have major concerns, while minor concrn is a lack of clear report of molecular mechanisms that connect the disease with symptoms and signs. Perhaps, simple and clear figure or schema of pathogenesis of Chronic Fatigue Syndrome in relation with other loneliness could increase an attractivity of the paper. I recommend to accept the article after minor revision.

Author Response

Thank you for your kind words about our piece. It is always nice to read such feedback.

As per your suggestion, we added a paragraph on the molecular mechanisms of the disease, and how it is connected to the symptoms.

Reviewer 3 Report

This article concerns a topic of high interest in public health fields. However, some observations are needed.

#Methods. The research methodology description is missing: it is a gap.

The article is undoubtedly a review, but the authors have not explained the type of article (narrative review, systematic review, scoping review?) or the research methodology. Therefore in the face of a vast number of bibliographic sources consulted and reported, the reader cannot understand how the search for sources took place.

A methods section, well structured and exhaustive in content, is mandatory.

#Results ad discussion

These two areas would appear to be reported at the same time. Even if it could be methodologically correct, this approach, along with the absence of summary tables and flowcharts, makes the text not very accessible to the reader.

Authors should take steps to encourage understanding and avoid redundancies.

In particular, paragraph 6 “Loneliness and illness” is dispersive and, while understanding the novelty of the issue linked to COVID 19, its discussion distracts from the ME / CFS theme.

The Results and Discussion section needs to be better harmonized.

Author Response

Regarding the passage you quoted- we had actually misquoted “(Friedberg et al., 2012)” instead of “(Friedman et al., 2019”. Neu et al., 2014 is a secondary source from Friedman, speaking about the psychosomatic features of the disorder. We revised the sentence for accuracy and thank you for pointing that out to us- as a precaution we have rechecked our references, and are now certain that the one that are left, belong there.

Author Response

Attached my response to reviewer #4

Round 2

Reviewer 3 Report

Please see the file attached

Author Response

I added a word document with my response to the reviewers.